# Recycling Waste Nonmetallic Printed Circuit Boards for Polyvinyl Chloride Composites

**DOI:** 10.3390/polym14173531

**Published:** 2022-08-28

**Authors:** Aung Kyaw Moe, Jirasuta Chungprempree, Jitima Preechawong, Pornsri Sapsrithong, Manit Nithitanakul

**Affiliations:** 1The Petroleum and Petrochemical College, Chulalongkorn University, Bangkok 10330, Thailand; 2Center of Excellence on Petrochemical and Materials Technology, Bangkok 10330, Thailand; 3Department of Mechanical Engineering Technology, College of Industrial Technology, King Mongkut’s University of Technology North Bangkok, Bangkok 10800, Thailand

**Keywords:** nonmetallic printed circuit board, silane coupling agent, polyvinyl chloride composite

## Abstract

To reduce environmental threats, such as land filling, incineration and soil pollution, which are associated with the improper waste management of waste printed circuit boards, the utilization of NMPCBs from waste PCBs as a filler in composites was pursued. Untreated and treated NMPCBs in varying ratios, 10–30 wt.%, were blended with PVC to produce NMPCB/PVC composites, using the melt-mixing method via an internal mixer, in order to solve the remaining NMPCB waste problem after the valuable metals in PCBs were recovered. The incorporation of the NMPCB with PVC resulted in an increase in the tensile modulus and the thermal stability of the resulting composites. Scanning electron microscopy (SEM) results indicated improved interfacial adhesion between the treated NMPCB and the PVC matrix. The FTIR results of the NMPCB treated with 3-glycidyloxypropyltrimethoxysilane (GPTMS) revealed the formation of Si-O-Si bonds. The densities of the composites were found to increase with an increase in the content of the treated NMPCB, and compatibility improved. The tensile properties of the treated NMPCB/PVC composites were higher than those of the untreated NMPCB/PVC composites, suggesting improved compatibility between the treated NMPCB and PVC. The PVC composite with 10 wt.% of the treated NMPCB showed the optimum tensile properties. It was observed that the tensile modulus of the treated NMPCB/PVC composite increased by 47.65% when compared to that of the neat PVC. The maximum thermal degradation temperature was 27 °C higher than that of the neat PVC. Dynamic mechanical analysis results also support the improved interfacial adhesion as a result of the improvement in the storage modulus at the glassy region, and the loss factor (tan δ) peak shifted to a higher temperature range than that of the PVC and the untreated NMPCB/PVC composite. These studies reveal that the NMPCB was successfully modified with 1 wt.% of GPTMS, which promoted the dispersion and interfacial adhesion in the PVC matrix, resulting in better tensile properties and better thermal stability of the PVC composite.

## 1. Introduction

In recent years, waste electrical and electronic equipment (WEEE) has significantly increased because of its short life span and the innovation of technology [1]. All electrical and electronic equipment includes printed circuit boards (PCBs), which are classified into metallic fractions (MFs) containing valuable metals and nonmetallic fractions (NMFs) containing thermoset resins, additives of flame retardants and other additives. PCBs are also classified as being single/double sided (commercially defined as FR-2 type) or multi-layered (commercially defined as FR-4 type), which are NMFs, with the former containing cellulose laminated phenolic resin and the latter containing multifunctional epoxy resins or cyanide resin [2]. Recently, the waste recycling of WEEE has become an important issue because of the lack of appropriate technologies and the risk of releasing hazardous and toxic substances if WEEE is deposited in landfills and incinerated. The recovery of valuable metals, such as Au, Ag, Cu, Pd and Pt, from WEEE is one of the driving forces of recycling. However, the thermoset resin of nonmetallic printed circuit boards (NMPCBs), with the rigid particle characteristic, cannot be recycled using traditional melting technology. Hence, several studies have focused on the application of NMPCBs in chemical recycling and mechanical recycling for the mitigation of environmental problems and fulfillment of new regulations [3,4]. Currently, waste recycling focuses on and takes into account the economic and environmental consequences, such as the reduced energy consumption (low-carbon footprint) and the cost of production [5], sustainable processes with eco-friendly products [6,7] and upcycling products with the use of renewable green solvent in the recycling process [8].

Nearly 70 wt.% of NMPCBs from waste PCBs contain thermoset resins and glass fibers. NMPCBs need to be reused to reduce resource waste and secondary pollution from disposal in landfills and combustion. Hence, the need for the study of a novel reuse method for the incorporation of NMPCBs in recycled polymers to improve the Young’s modulus and flexural modulus of polymer composites because of the rigid nature and glass fibers of NMPCBs. Furthermore, it has been shown that NMPCBs can improve the thermal stability and flame-retardant properties of polymer composites [9,10,11]. However, the incorporation of nonmetallic printed circuit board (NMPCB, FR-4 type) powder into polymers (recycled high-density polyethylene, polypropylene and recycled acrylonitrile butadiene styrene) has been shown to cause poor interfacial adhesion between the polymer matrix and the filler, resulting in the composites having poor mechanical properties. In a previous study, in order to improve the compatibility (interfacial adhesion) of NMPCB/polymer composites, a silane coupling agent (ATPS) and a compatibilizer (maleic anhydride grafted polyolefin or ABS) were employed. Two different grades and types of recycled PVC (RPVC), waste wires and cables, and waste pipes were used as the matrix to make RNMPCB/PVC composites with different methods of press molding, solvent casting and melt compound. It was concluded that the mechanical properties of the resulting RNMPCB/PVC composites were governed by the grade and type of the RPVC compound, the size and contents of the NMPCBs, and the processing methods [12,13,14]. Furthermore, the product quality of the RPVC product deteriorated during the recycling process, and the service life is a drawback of using recycled PVC for industrial applications [15,16].

The objective of this study is the direct reutilization of NMPCBs from waste PCBs in PVC composites through the use of mechanical recycling methods for the potential application of profiles to help reduce production cost, solve environmental threats (land filling, soil pollution and open burning) and prevent resource depletion as a means to support the circular economy with an improvement in polymer composite properties. Furthermore, the renewable source of epoxidized soybean oil (ESBO), as a bio-based plasticizer, was employed in this study to produce a PVC composite that is more economical and environmentally friendly, with the aim to persuade the public to use PVC composites made from recycled NMPCBs instead of petroleum-based phthalate plasticizers.

To the best of our knowledge, no study has been carried out on the effect of untreated and silane-coupling-agent-treated nonmetallic printed circuit board powder (NMPCB) (FR-4 type) as a filler in bio-based plasticized PVC composite for a novel reuse method for NMPCBs. The aim of this work was to investigate the effects of NMPCBs from waste PCBs and 1 wt.% 3-glycidyloxypropyltrimethoxysilane (GPTMS)-treated NMPCBs on the mechanical properties of PVC composites. The effect of the addition of untreated and treated NMPCBs into PVC on the thermal degradation of PVC composites was also examined. The interfacial adhesion or interaction of the NMPCB in the PVC matrix was studied using a dynamic mechanical analyzer and density measurement. An SEM-EDX analyzer was used in the studies of the dispersion and adhesion of the NMPCB in the PVC matrix.

## 2. Materials and Methods

### 2.1. Materials

Polyvinyl chloride (PVC) of suspension grade No. SG660, with a K value of 66, an apparent bulk density of 0.55 g mL^−1^ and an impurity of less than 5%, was used as the polymer matrix, and it was purchased from Thailand Plastic and Chemical company (TPC, Bangkok, Thailand). Several additives were added to the PVC powder in order to improve the melt processing before compounding. The additives in the formulation of the PVC compound were calcium stearate (632.33 g mol^−1^ molecular weight, Sigma-Aldrich, Taufkirchen, Germany) and zinc stearate (607.02 g mol^−1^ molecular weight, Sigma-Aldrich, Taufkirchen, Germany) as thermal stabilizers and epoxidized soybean oil (ESBO, Chemmin Company, Bangkok, Thailand) as a bio-based plasticizer. The physical characteristics of ESBO were 6.5 g mL^−1^ and oxirane functionality 6%. The coupling agent 3-glycidyloxypropyltrimethoxysilane (GPTMS, molecular weight of 236.34 g mol^−1^, density of 1.07 g mL^−1^ at 25 °C, purity ≥ 98%) was obtained from Sigma-Aldrich (Taufkirchen, Germany).

The recycled nonmetallic printed circuit board (NMPCB, FR-4 type) powder used in the study was waste from the trimming of the edge of a printed circuit board, which was from post-industrial waste. This NMPCB powder of the waste PCB consisted of a woven fiberglass mat impregnated with thermoset resins (epoxy resin or phenolic resin etc.). The NMPCB powder, used as a filler in the PVC composite, was attained from passing the NMPCB powder through a sieve with an aperture of 180 μm (ASTM mesh No. 80 sieve).

### 2.2. Surface Modification of Recycled NMPCB Powder

The surface of the recycled NMPCB powder was modified with 1 wt.% content of the silane coupling agent (3-glycidyloxypropyltrimethoxysilane, GPTMS) for better interfacial adhesion between the PVC matrix and the surface of the NMPCB by means of silanization for 1 h at 80 °C under agitation and reflux conditions. Prior to the silanization process, the silane coupling agent solution was developed using 1 wt.% of GPTMS (based on the weight of NMPCB) mixed with ethanol solution (ethanol-deionized water, volume ratio 70:30) for 30 min at 500 rpm with a magnetic stirrer. The NMPCB suspension was then filtered. The surface-treated NMPCB powder was further dried in an oven at 70 °C for 24 h before subsequently being used in the NMPCB/PVC composites.

### 2.3. Composite Compounding

The PVC composites were compounded with 30 phr of ESBO as a plasticizer and 1 phr of Ca/Zn stearate based on 100 phr of PVC resin, with the composition of the NMPCB varying from 10 to 30 wt.%, in a Hake Rheomix at 150 °C with a rotor speed of 60 rpm for 6 min. After thoroughly mixing, the compound was removed from the Hake Rheomix. The compound was transferred to a mold with dimensions of 160 mm × 160 mm × 3 mm (length, width and thickness) under an electrically heated hydraulic press machine (Lab Tech, Samutprakarn, Thailand). The compound was then preheated for 10 min at 165 °C, followed by hot pressing at the same temperature for another 10 min. Finally, cooling was carried out for 10 min under pressure.

### 2.4. Characterization of NMPCB/PVC Composites

The recycled NMPCB powders were characterized by Fourier transform infrared (FTIR) spectroscopy (Perkin Elmer Spectrum One) to identify the functional groups and to confirm the surface modification of the recycled NMPCB powder. The mechanical properties of the NMPCB/PVC composites with varying amounts of untreated and treated NMPCBs were measured using a Lloyd universal testing machine (type LDX, West Sussex, UK), according to ASTM D882. Rectangular samples with dimensions of 120 mm × 10 mm × 0.8 mm were cut from a compression sheet. The gauge length was set to 50 mm at a crosshead speed of 50 mm min^−1^. The results of the tensile test were obtained by averaging the results of five specimens. A dynamic mechanical analysis of the plasticized PVC, and the untreated and treated NMPCB/PVC composites with dimensions of 50 mm × 3 mm × 3 mm (length, width and thickness) was performed using a DMA (GABO, EplexorR, type 100 N, Ahlden, Germany) with the tensile mode of the equipment. The viscoelastic properties of the samples were determined under the conditions of the temperature ranging between 10 °C and 140 °C, with a heating rate of 1 °C min^−1^, under nitrogen flow and at a fixed frequency of 1 Hz of static strain of 0.2 %. The thermal degradation of the NMPCB/PVC composites was evaluated using a thermogravimetric analyzer Perkin-Elmer Pyris Diamond TG-DTG instrument (model RIS diamond TG-DTG, high temp 11) on 6 mg samples over a temperature range between 30 °C and 800 °C, at a heating rate of 10 °C min^−1^ and under a nitrogen flow of 20 mL min^−1^. A scanning electron microscope (SEM) (JEOL, Hitachi, Model S4800, Tokyo, Japan) was employed to study the interfacial morphology of the untreated and treated NMPCB/ESBO plasticized PVC composites; the test specimens were sputter-coated with a thin layer of gold to prevent electrical charging during the observation. SEM/EDX was also used to confirm the chemical compositions of the waste NMPCBs in the PVC composites. Density measurements were performed at room temperature according to ASTM D792 on a Sartorius balance by means of the Archimedean principle (water displacement method). Distilled water was used as the liquid. For each sample, the average value of three measurements was reported.

## 3. Results and Discussion

### 3.1. Characterization of Untreated and Treated NMPCBs

The SEM micrographs of the untreated NMPCB and treated NMPCB are presented in Figure 1a,b respectively. As shown in Figure 1, both the untreated and treated NMPCBs contained irregular thermoset resin powders, and glass fibers of various lengths and sizes were also observed; the results are consistent with those of Zheng Y. et al. [17]. The smooth surface of the treated NMPCB can be observed in Figure 1b. For the surface characterization, FTIR spectroscopy was successively applied to determine the functional changes between the surfaces of the untreated NMPCB and treated NMPCB.

Figure 2 exhibits the FTIR spectra of GPTMS, the untreated NMPCB and the treated NMPCB from 600 to 4000 cm^−^^1^. The FTIR spectrum of GPTMS revealed peaks at 1190, 1074, 909 and 815 cm^−^^1^, which were related to the stretching of CH_3_, Si-O-CH_3_ (stretching vibration of ethoxy groups directly bonded to the silicon atom), C–O (stretching of the oxirane group) and Si-O-C bonds, respectively [18,19], as shown in Figure 2. After the surface modification of the NMPCB, as shown in Figure 2, the treated NMPCB shows a sharp peak at 2935 and 2869 cm^−^^1^, which was attributed to the stretching peak of alkyl-CH, -CH_2_ and –CH_3_ from GPTMS. In addition, the presence of a broad peak between 3700 cm^−^^1^ and 3000 cm^−^^1^ suggests that there were hydrogen bonds on the surface of the treated NMPCB due to the hydroxyl groups of silanol (hydrolysis reaction of GPTMS during the silanization process) and the polar functional groups of the NMPCB. This fact supports the notion that there were interactions between the modified waste NMPCB and GPTMS. The characteristic peaks between 1800 cm^−^^1^ and 650 cm^−^^1^ could not be analyzed, probably because of the overlapping peaks of the spectra of the untreated NMPCB as a result of the complex components in the NMPCB. The FTIR spectrum of the NMPCB revealed sharp peaks at 1011, 929 and 822 cm^−^^1^, which correspond to Si-O-Si and Si-O functional groups. These peaks confirm the evidence of the existence of glass fibers in the NMPCB of the waste PCB. The peaks at 1236 and 1179 cm^−^^1^ correspond to C-O-C bonds, confirming the presence of an epoxy matrix in the NMPCB of the waste PCB.

### 3.2. Characterization of NMPCB/PVC Composites

#### 3.2.1. Morphology of NMPCB/PVC Composites

The scanning electron micrographs of the cross-section surface of the PVC and the NMPCB/PVC composites with varying amounts of untreated NMPCB content from 10 wt.% to 30 wt.% are shown in Figure 3a–d. For neat PVC, the SEM micrograph shows a relatively smooth surface in Figure 3a. With a higher content of the untreated NMPCB, much thermoset resin and glass fiber pullout from the PVC matrix could easily be visible during the fracture, as shown in Figure 3b–d. It was also observed that the cross-section surface of the 20 wt.% untreated NMPCB/PVC composite and the 30 wt.% untreated NMPCB/PVC composite revealed a distinct gap between the irregular-shaped thermoset resin and the PVC matrix and glass fiber pullout. This indicates poor interfacial adhesion between the NMPCB and the PVC matrix, as shown in Figure 3c,d.

On the contrary, the SEM observation of the NMPCB/PVC composite with the treated NMPCB indicated that the glass fibers and thermoset resin were well dispersed in the PVC matrix, as shown in Figure 3e–g. The glass fibers were covered with a layer of matrix linking to the PVC matrix. Therefore, the interaction between the treated NMPCB and the PVC matrix improved, as observed in Figure 3e–g. This was associated with the interaction between the epoxy functional group of the treated NMPCB and the PVC molecular chains via the ring-opening reaction during melt compounding because the GPTMS-treated NMPCB contained epoxy functional groups. Badra Bouchareb et al. revealed the possible ring-opening reaction of the epoxy functional groups in ESBO and PVC molecular chains, resulting in ether bond formation [20]. Furthermore, Jie Chen demonstrated that the epoxy functional groups of the epoxidized glycidyl ester of the ricinoleic acetic ester reacted with PVC molecular chains via the ring-opening reaction [21]. Lanjun Li studied the crosslinking of rigid PVC with GTPS. They found the formation of ether linkage, which resulted from the epoxy groups of GTPS-substituted allylic chlorines on PVC molecular chains [22,23]. Therefore, the SEM analysis showed improved interfacial adhesion between the treated NMPCB and the PVC matrix.

Subsequently, an EDX analysis was performed to verify the main component of the glass fibers (Si) and the dispersion of the glass fibers in the NMPCB/PVC composites. When increasing the amount of NMPCB from 10 wt.%, 20 wt.% and 30 wt.%, the EDX results showed that the silicon element content increased, as presented in Figure 4, and the EDX image also indicated a good dispersion of Si in the treated NMPCB/PVC composites.

#### 3.2.2. Density

The PVC and NMPCB/PVC composites were subjected to a density measurement, and the densities of the samples are given in Table 1, which shows that the densities of the composites depended on the amount of NMPCB filler loading in PVC since the density of an NMPCB is higher than that of PVC. Moreover, the higher density of the NMPCB/PVC composites containing treated NMPCB reflected the low interfacial gap (void) formation between the treated NMPCB and the PVC matrix as a result of improved interaction compared to the untreated NMPCB/PVC composites.

#### 3.2.3. Mechanical Properties of NMPCB/PVC Composites

The untreated and treated NMPCBs were used to prepare NMPCB/PVC composites with the melt compounding method. Figure 5 shows the Young’s modulus of the PVC, and the untreated and treated NMPCB/PVC composites as a function of NMPCB content. It was observed that the Young’s modulus of both the untreated NMPCB/PVC composites and the treated NMPCB/PVC composites increased with an increase in the amount of NMPCB content when compared with that of the PVC. This was due to the fact that the NMPCB is more rigid than the PVC matrix. The Young’s modulus of a composite depends on the following factors: (i) the modulus of the filler, (ii) a high aspect ratio and (iii) the filler contents [24]. This result is typically observed in thermoplastic composites due to the incorporation of rigid filler to viscoelastic matrices that hinder polymer chain mobility [25,26,27]. The incorporation of the NMPCB filler (both untreated and treated) in the NMPCB/PVC composites, which had a high modulus, gave rise to a more rigid material. Compared with the untreated NMPCB/PVC composites, the Young’s modulus of the NMPCB/PVC composites filled with 10 wt.%, 20 wt.% and 30 wt.% of the treated NMPCB increased from 332 MPa, 394 MPa and 400 MPa to 378 MPa, 475 MPa and 497 MPa, respectively. The incompatibility of the untreated NMPCB was expected to cause the NMPCB/PVC composite to have poorer mechanical properties. The improved treated NMPCB/PVC composite was attributed to the fact that GPTMS- or silane-modified surface of the NMPCB enhanced the interfacial adhesion and, thus, improved the interfacial bonding between the NMPCB and the PVC matrix.

Figure 6 presents the tensile strength of the neat PVC, and the untreated and treated NMPCB/PVC composites as a function of NMPCB content. It was clearly observed that increasing the content of the untreated NMPCB in the PVC composites was the cause of the decrease in the tensile strength due to the fact that the increase in the amount of glass fibers in the NMPCB filler hindered the flowability of polymer chains and caused a poor dispersion of filler in the matrix [28]. This result was also observed by Muniyandi et al. [29,30]. Rajesha K. Das et al. found that, in PVC composites with varying amounts of NMPCB content from 0 to 10 wt.%, the weak interaction between the NMPCB and recycled PVC (r-Wire cable) resulted in a decline in the tensile strength as the NMPCB content was increased [14]. The tensile strength of the PVC composite filled with various contents of untreated NMPCB from 10 wt.% to 30 wt.% was due to the inability of the NMPCB filler to support the stress transferred from the PVC matrix. When compared with the untreated NMPCB/PVC composites, the tensile strength of the GPTMS-treated NMPCB/PVC composites increased by 24.4%, 28.6% and 23.3%. This was due to the increase in the interfacial adhesion caused by the surface treatment of GPTMS on the NMPCB. It was clearly observed that 1 wt.% GPTMS treatment caused the tensile strength of the treated NMPCB/PVC composite with 10 wt.% of the treated NMPCB content to be rather similar to that of the PVC. However, with 20 wt.% of treated NMPCB content, the tensile strength of the PVC composites decreased by 17.7%, and with 30 wt.%, it decreased by 15.2% when compared to the tensile strength of the PVC. This is likely to be associated with the self-interaction or agglomeration of the treated NMPCB particles between itself when the NMPCB content was increased.

#### 3.2.4. Thermal Property

The thermal stability of NMPCB/PVC composites is important not only for processing purposes but also for the application of the final products. The thermogravimetric analysis (TGA) and derivative thermogravimetric analysis (DTGA) curves of the PVC, waste NMPCB powder and NMPCB/PVC composites with varying contents of NMPCB are shown in Figure 7 and Figure 8. The PVC and NMPCB/PVC composites showed a two-step thermal degradation. The first-stage degradation temperature ranging between 230 and 396 °C was due to the degradation of the plasticizer and the dehydrochlorination of the PVC forming an olefinic double bond, and the second-stage degradation temperature ranging between 396 and 538 °C was attributed to the degradation of the PVC chains of the PVC after dichlorination (the cracking of the conjugated polyene) [31,32,33,34]. In this study, the thermogram of the NMPCB revealed only one-step degradation at a temperature range between 200 °C and 470 °C, with a peak at 356.20 °C and a weight loss of 31.69%, which are consistent with the degradation of epoxy resin found in waste NMPCBs. Subsequently, the PVC compound and the NMPCB/PVC composites also revealed a two-step degradation, as shown in Figure 8. This indicates that the NMPCB/PVC composites, which predominantly consisted of PVC, showed similar characteristics to the PVC. Based on the DTGA curves, the degradation temperatures at the maximum peaks were higher in the NMPCB/PVC composites with different contents of incorporated NMPCB than in the PVC. This was due to the thermal stability of NMPCB being higher than that of the PVC compound. Therefore, the NMPCB possessed a higher thermal stability and presumably acted as the protective layer to reduce the permeability of the volatile HCl gas (dehydrochlorination reaction), which was produced from the thermal decomposition of the PVC composites. The improved thermal stability might also be caused by the interaction between the polar functional groups of the NMPCB and HCl evolved from the PVC matrix during the decomposition of the NMPCB/PVC composite. It is interesting to note that the incorporation of the NMPCB in the PVC composite was significantly effective in the first-stage degradation because interactions between the NMPCB and the volatile gases from the PVC mainly occurred. Figure 8 shows that the incorporation of 10 wt.% NMPCB significantly improved the thermal stability of the NMPCB/PVC composite. From this study, it was found that the peak in the first-stage degradation temperature of the treated NMPCB/PVC composites with higher contents of treated NMPCB (20 wt.% and 30 wt.%) shifted to a higher temperature when compared with that of the untreated NMPCB/PVC composites. This could be explained by the effect of the GPTMS silane coupling agent, which helped to improve the NMPCB–PVC matrix interaction in the treated NMPCB/PVC composites. The char yields of the PVC composites were dependent on the contents of the NMPCB, which consisted of glass fibers.

#### 3.2.5. Dynamic Mechanical Analysis

The intrinsic damping properties (dynamic mechanical properties) of the composite is a crucial and key interest for the applications of the material in industry. Dynamic mechanical analysis is a particularly convenient method to access the damping properties (the loss factor (tan δ) or damping factor, the storage modulus and the loss modulus) of composites. The dynamic mechanical properties of composites depend on various factors: fiber loading, the orientation of the fiber and the nature of the fiber–matrix interface region [25,26,27]. The present research concentrated on the effect of NMPCB loading and the interfacial region. The variations in the storage modulus (E′) and the damping factor as a function of temperature are reported in Figure 9a,b respectively.

In Figure 9a, the increase in the storage modulus (E′) at the glassy region (room temperature) occurred with an increase in the amount of untreated NMPCB loading of 10, 20 and 30 wt.% in the NMPCB/PVC composites compared to the PVC. This was due to the modulus of elasticity (Young’s modulus) of the NMPCB of the filler being greater than that of the PVC. Above the rubbery region (T_g_), the storage modulus (E′) of the NMPCB/PVC composites did not show any obvious differences because of the softening of the polymer matrix at a high temperature or the plasticizing effect of the bio-based plasticizer ESBO in the polymer matrix. It is interesting to note that the NMPCB powder in the PVC composite provided a higher storage modulus or stiffness at room temperature.

For the treated NMPCB/PVC composites, the obtained storage modulus (E′) was higher than that of the untreated PVC composites. This result is due to the stronger interfacial adhesion between the treated NMPCB and the polymer matrix, resulting in an improved stress transfer between the NMPCB and the polymer matrix. The maximum E′ value, which was 1372.38 MPa, was exhibited by the treated NMPCB/PVC composite with 30 wt.% of NMPCB loading, whereas the E′ value of the neat PVC was 932.57 MPa at room temperature. It was observed that the E′ value of the treated NMPCB/PVC composite loading at room temperature was improved by 47.16% upon the incorporation of 30 wt.% of the treated NMPCB. Above the rubbery region (temperature range after T_g_), all untreated and treated NMPCB/PVC composites did not show any significant changes in E′ values, and the influence of the NMPCB compositions on the stiffening of the PVC matrix was marginal. This demonstrates that the interfacial interaction was significantly imparted on the storage module at a temperature lower than the glass transition temperature because of freezing and warping between the polymer matrix chain and filler. However, the occurrence of the mobility and deformation of the polymer chain increased with an increase in the temperature under the action of external forces. As a consequence, the weakening of the interfacial adhesion caused the diminishing of the storage modulus [35]. Therefore, the E′ values of the NMPCB/PVC composites with different compositions of the untreated and treated NMPCB filler at the rubbery region (temperature range after T_g_) were similar to that of the ESBO plasticized PVC due to the mobility of the molecular chains in the NMPCB/PVC composites becoming more intense under external forces, which consequently decreased the interfacial adhesion. Identical storage modulus behavior of treated SiO_2_/PVC composites was also reported by Chandra R. et al. [36].

The damping factor can be expressed by the ratio of the loss modulus (E″) to the storage modulus (E′) and is defined as tan δ. This ratio (tan δ) depends on the filler and the polymer matrix interaction level. The tan δ_max_ peak is associated with the information of the glass translation temperature (T_g_) and the energy dissipation of composite materials. In Figure 9b, it can be observed that the value of the tan δ peak decreased with an increase in the loading of the untreated NMPCB in the PVC from 0.42 to 0.39 in comparison to the peak value of tan δ (0.42) in the plasticized PVC compound. This indicates that the addition of the NMPCB hindered the mobility of the PVC molecular chains, resulting in a lower flexibility and lower molecular mobility. This result is in agreement with surface-treated waste cellulose composites [37,38,39]. Another factor for the lower value of the tan δ peak related to the larger amount of NMPCB filler occurred when filler–filler contact (agglomeration) and phase separation increased; hence, the filler–matrix interaction decreased in the small volume of the polymer matrix, leading to less energy dissipation. Therefore, the untreated NMPCB showed the characteristic of a low value of tan δ.

When the treated NMPCB was incorporated into the PVC matrix, the tan δ_max_ peak values increased sharply. Chen G. et al. suggested that an increased loss factor (tan δ_max_) occurred by means of an increase in the damping force of the molecular chain motion because of the increase in the contact of the PVC molecular chains and the effective volume fraction of the filler (silane-treated SiO_2_) [40]. Chen G. et al. concluded that there was a stronger interfacial interaction between the filler and the polymer matrix, a larger volume fraction of the filler and a higher loss factor (tan δ_max_ peak intensity) of the composite. Sreenivasan V.S. et al., studying the dynamic mechanical properties of sansevieria cylindrica/polyester composites, reported that the effectiveness of the interfacial adhesion at the treated fiber/matrix interface was due to an enhancement in the magnitude of the tan δ_max_ peak, which was linked to interfacial thickness, the bending of fibers, the breaking of fibers, matrix cracking and the formation of fiber pullout [39].

## 4. Conclusions

Untreated and treated NMPCB/PVC composites were prepared with different contents of NMPCB powder from a waste PCB by using melting mixing. The SEM morphology of the treated NMPCB/PVC composites showed a good dispersion of the NMPCB powders in the PVC matrix. When the amount of the NMPCB increased from 10 to 30 wt.%, the surface became rough as can be seen in the SEM images. The properties of the NMPCB/PVC composites revealed that the tensile modulus increased continuously as the content of the NMPCB increased in both the untreated and treated NMPCB/PVC composites due to the addition of reinforcing fillers. The maximum tensile modulus of 497 MPa was achieved with 30 wt.% of the treated NMPCB/PVC composite. Furthermore, the incorporation of the NMPCB enhanced the thermal stability of the NMPCB/PVC composites, and the T_d_ values of the NMPCB/PVC composites shifted from ~290 °C for the PCV to ~300 °C for the NMPCB/PVC composites. The addition of the NMPCB increased the density from ~1.23 g cm^−3^ of the PVC up to 1.35 and 1.36 g cm^−3^ of 30 wt.% of the untreated and treated NMPCB/PVC composites, respectively. The dynamic mechanical analysis also confirmed these observations, showing an increase of 47.16% in the storage modulus and an increase of 29.83% in the peak intensity of tan δ_max_ (loss factor) in the PVC composite with the incorporation of 30 wt.% of the treated NMPCB. It could be concluded that the GPTMS coupling agent effectively improved the dispersion of the NMPCB in the PVC matrix and the interaction between the treated NMPCB and the PVC matrix with less void formation, providing a unique opportunity for the production of NMPCB/PVC composites to obtain both thermal stability and satisfactory mechanical properties as a second-generation composite with value-added high-performance applications.

## Figures and Tables

**Figure 1 polymers-14-03531-f001:**
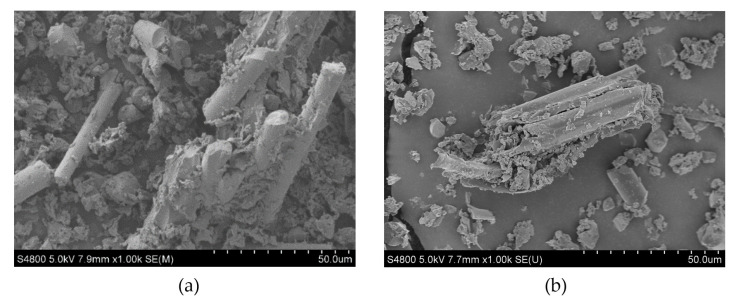
SEM image of nonmetallic printed circuit board (NMPCB) powder: (**a**) untreated and (**b**) treated.

**Figure 2 polymers-14-03531-f002:**
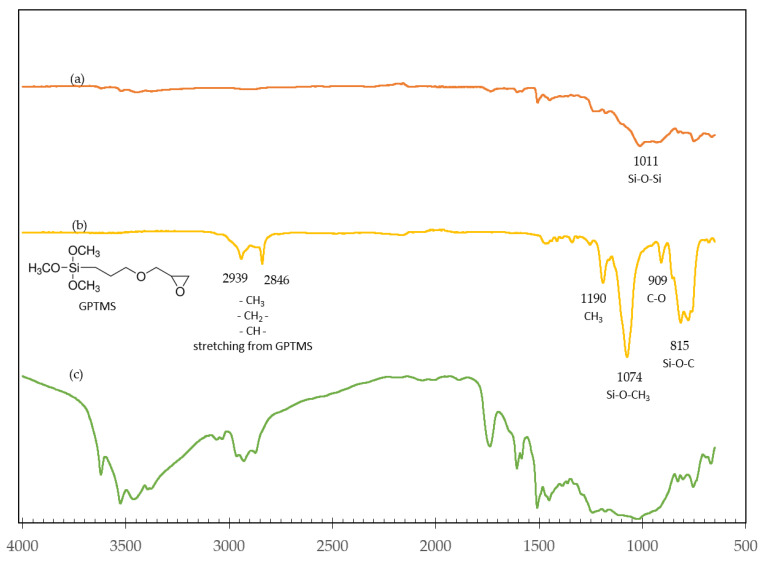
FTIR of (**a**) untreated NMPCB; (**b**) 3-glycidyloxypropyltrimethoxysilane (GPTMS); (**c**) treated NMPCB.

**Figure 3 polymers-14-03531-f003:**
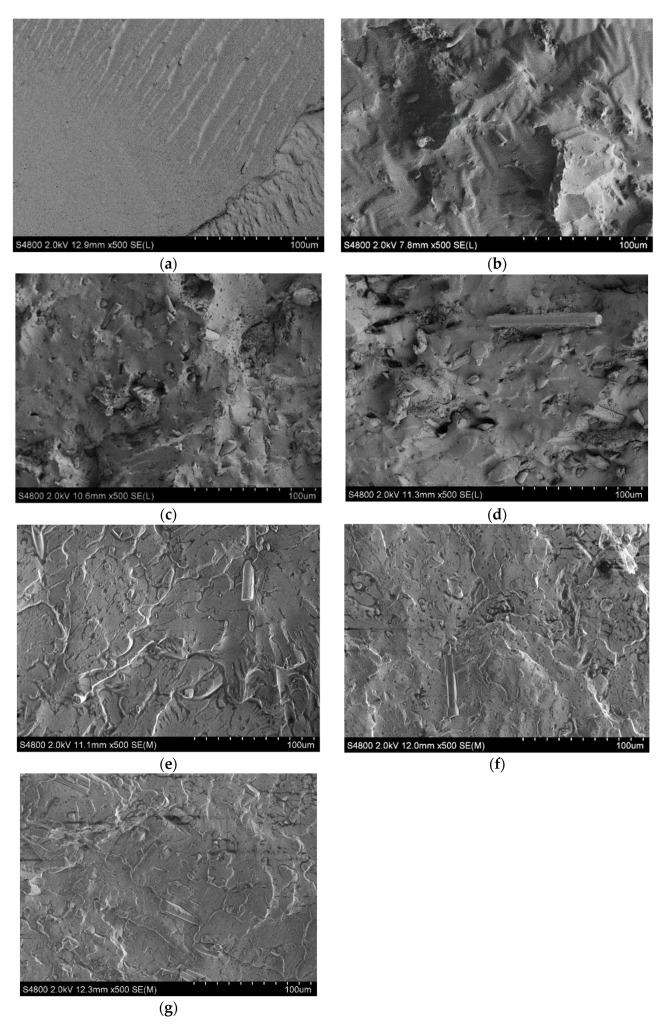
SEM morphology of (**a**) neat PVC, (**b**) 10 wt.% untreated NMPCB/PVC composite, (**c**) 20 wt.% untreated NMPCB/PVC composite, (**d**) 30 wt.% untreated NMPCB/PVC composite, (**e**) 10 wt.% treated NMPCB/PVC composite, (**f**) 20 wt.% treated NMPCB/PVC composite and (**g**) 30 wt.% treated NMPCB/PVC composite.

**Figure 4 polymers-14-03531-f004:**
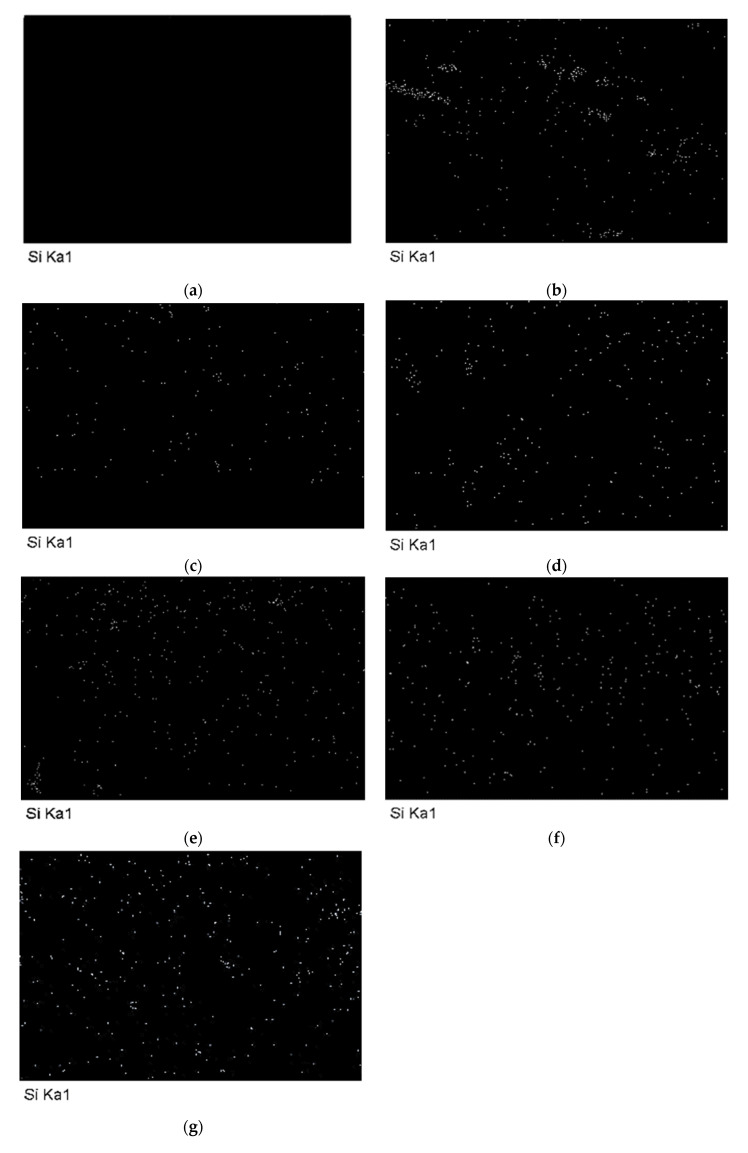
Silicon element on EDX mapping image of (**a**) neat PVC, (**b**) 10 wt.% untreated NMPCB/PVC composite, (**c**) 20 wt.% untreated NMPCB/PVC composite, (**d**) 30 wt.% untreated NMPCB/PVC composite, (**e**) 10 wt.% treated NMPCB/PVC composite, (**f**) 20 wt.% treated NMPCB/PVC composite and (**g**) 30 wt.% treated NMPCB/PVC composite.

**Figure 5 polymers-14-03531-f005:**
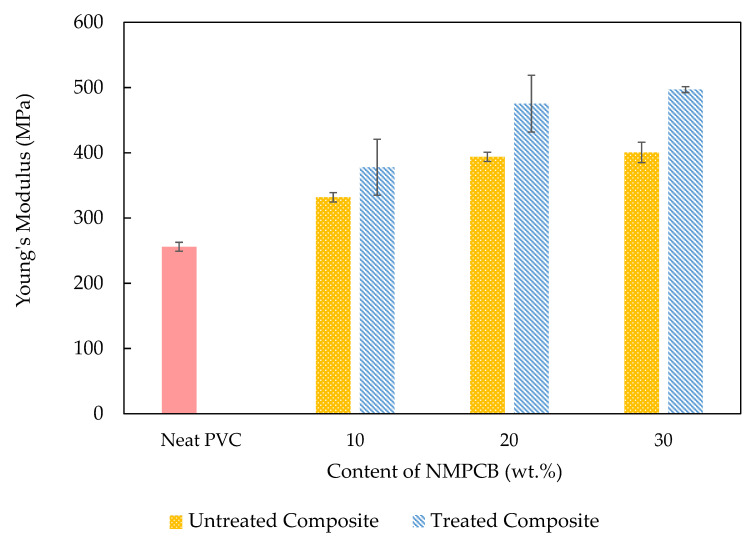
Young’s Modulus of untreated and treated NMPCB/PVC composites.

**Figure 6 polymers-14-03531-f006:**
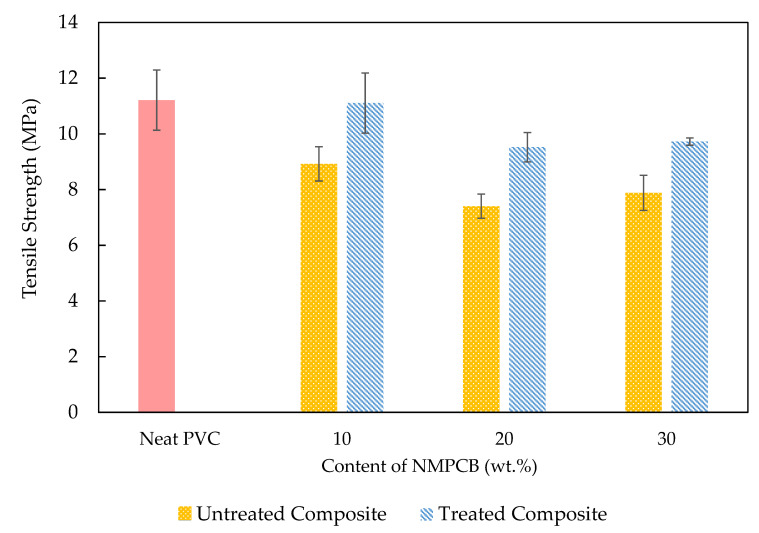
Tensile strength of untreated and treated NMPCB/PVC composites.

**Figure 7 polymers-14-03531-f007:**
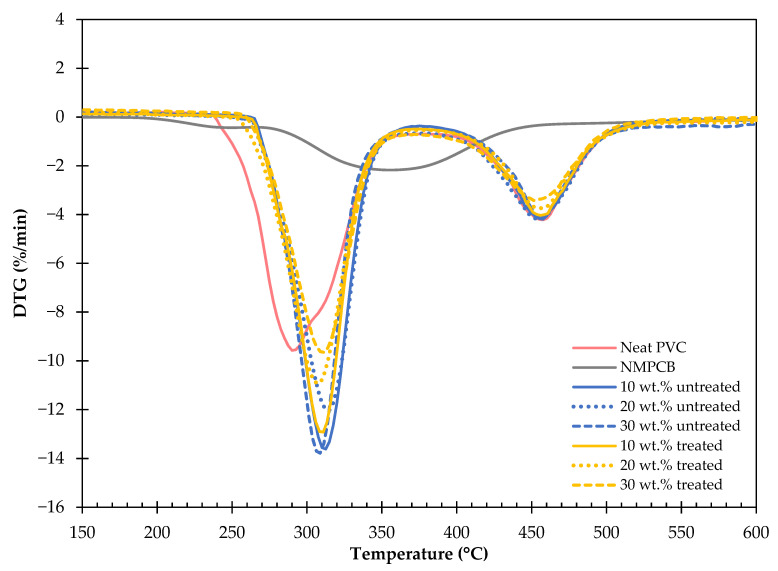
DTG of the untreated and treated NMPCB/PVC composites.

**Figure 8 polymers-14-03531-f008:**
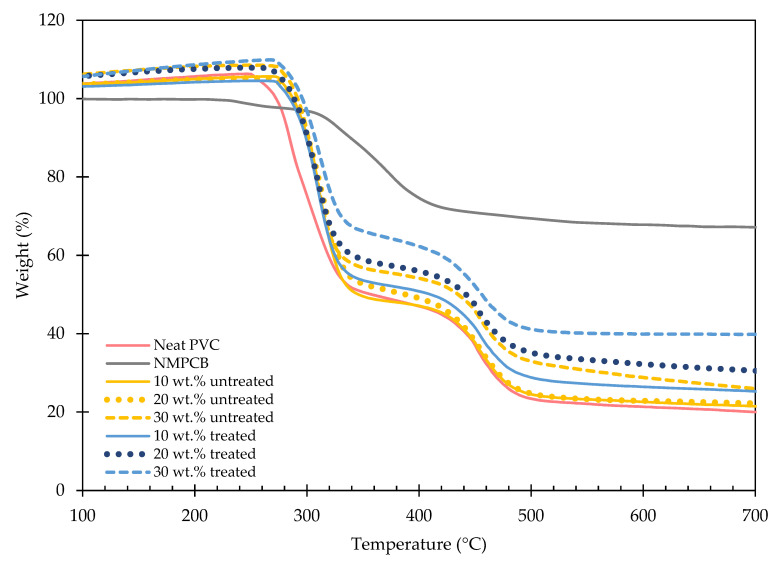
Weight loss of the untreated and treated NMPCB/PVC composites.

**Figure 9 polymers-14-03531-f009:**
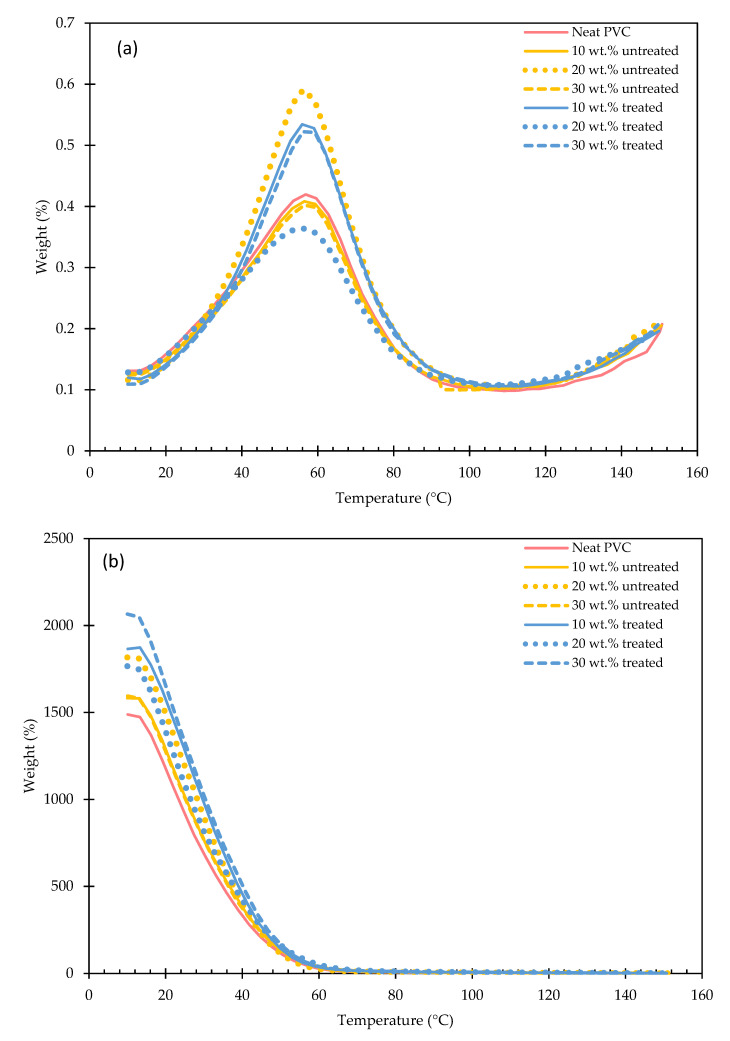
DMA of untreated and treated NMPCB/PVC composites: (**a**) storage modulus and (**b**) tan δ.

**Table 1 polymers-14-03531-t001:** Density of untreated and treated NMPCB/PVC composites.

Samples	Density (g cm^−3^)	
	Untreated	Treated
Neat PVC	1.23 ± 0.013	
10 wt.% NMPCB/PVC composite	1.31 ± 0.002	1.32 ± 0.001
20 wt.% NMPCB/PVC composite	1.34 ± 0.001	1.35 ± 0.005
30 wt.% NMPCB/PVC composite	1.35 ± 0.001	1.36 ± 0.005

## Data Availability

The authors confirm that the data supporting in this study are available within the article. Raw data that support the findings of this study are available from the corresponding author, upon reasonable request.

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
