# Peer review of "Recycling Waste Nonmetallic Printed Circuit Boards for Polyvinyl Chloride Composites"

_polymers, 2022, doi:10.3390/polym14173531_

Round 1

Reviewer 1 Report

1. The introduction does not discuss the state-of-the-art WEEE approaches. It should be corrected.

2. In addition to the SEM of the untreated and treated samples, EDX should also be performed and shown to reveal the composition.

3. How does the source of the waste material affect the results? Some waste materials have a large variance depending on their source. The authors should discuss this issue in the manuscript.

4. Some errors are reported. Are those standard deviations? However, it is unclear how these were derived and how many independently prepared materials were measured. Elaborate on this more, and give all the details on the sample preparation and repeated experiments.

5. The manuscript starts in medias res with WEEE. The importance of ‘waste recycling’ in general should be mentioned briefly, and recent emerging broad examples listed (10.1039/D1GC04709D; 10.1016/j.susmat.2022.e00448; 10.1039/D1GC03410C; 10.1021/acsapm.1c01918).

6. Both the quotient (“x/y”) and negative exponent (“x y-1”) formats are used in the manuscript for units. Either of them should be used consistently, preferably the negative exponent format, which is recommended by the IUPAC.

7. The experimental section should have a material section dedicated to list all of them with supplier, purity and grade, and all relevant information for reproduction of the work. This information is currently scattered and incomplete. Most information, e.g. purity, are missing.

8. The conclusion section is too short and vague. A good amount of results are presented, and compared to that only some of them made it into the conclusion section. The main research findings should be briefly summarized in quantitative statements.

Reviewer 2 Report

The paper seeks to introduce an approach ‘’ Recycling Waste Nonmetallic Printed Circuit Board for Polyvinyl Chloride Composite”. However, the authors should consider improving upon the quality to further highlight and emphasize.

1.    Based on the understanding of what an abstract should entail, consider one or two lines introducing the problem this study is trying to solve.

2.    Also, add a sentence highlighting the significance of the study at the end of the abstract.

3.    Put space between any variable and its respective unit. For instance, instead of 47.65 %, you represented it as 47.65% in line 28 which is not acceptable.

4.    The maximum number of words allowed in the keywords section is 3 but from the article, you have four (4) words. Consider reducing it to three words maximum.

5.    The introduction needs to be improved by relating to the mechanics of the studied materials and their mechanical characteristics. The references to be included are: 10.3390/polym14132662, 10.1016/j.jiec.2022.06.023, 10.1002/app.46770, 10.1016/j.compstruct.2021.114698 and 10.1016/j.polymertesting.2017.09.009.

6.    Your SEM analysis lacks many scientific data. For instance, in taking SEM analysis, information such as accelerating voltage, scale bar used, and the working range of the SEM device. Consider including these pieces of information.

7.    In section 3.1, modify the word “spectorscopy” to “spectroscopy.

8.    The SEM explanations does not justify anything. It was just an observation. The author needs to tell us why the results observed looks as it is. What functional groups prompted the results seen in the figure?

9.    Increase the font size of figure 2 and make all the figures unified.

10.The magnification footer of all the SEM analyses should be manually imputed insider the images for visibility’s sake.

Reviewer 3 Report

The resource utilization of waste circuit boards has always been a research hotspot in the industry. In this work, polyvinyl chloride (PVC) and modified surface waste non-metallic printed circuit board (NMPCB) was blending to prepare composite in order to find a new route of resource utilization of waste circuit boards. On the surface, this work seems very meaningful. In fact, this work lacks innovation in polymer blends. I do not think it can be published in polymers.

(1) Silane coupling agent was used to modify the filler to improve the interfacial compatibility between filler and polymer. This method is commonly used in polymer blends. Therefore, the technical route of this work is lack of innovation.

(2) The figures in the manuscript are confusing, and look like screenshots rather than real data charts.

(3) I believe that simple mechanical properties and rheological properties cannot fully characterize the properties of the composites.

(4) How the properties of the composites change when the NMPCB content is higher than 30%? This problem seems to be very important, because the properties of composites do not reach the inflection point when the addition amount is 30%.

To sum up, I think this work is not innovative enough, the workload is not enough, and the depth of discussion is not enough. Therefore, this manuscript must be rejected for publication in polymers.

Reviewer 4 Report

This is an interesting paper that investigates nonmetallic waste from printed circuit boards for use in PVC composites. Recycling of waste of printed circuit boards is of high interest, especially with the increasing attention for circular economy, recycling and avoiding waste.

The authors have used this waste or recycled powder, then applied a conventional surface modification method by use of silane coupling agent, and then compounded the PVC composites and characterized this material. 

The methodology is conventional and not very new. However, the specific application of using such recycled waste is of interest.

A major concern of this paper is the poor English language, due to which the meaning of several sentences is not clear, and this challenges understanding the discussion. Moreover, a space should be put just before the citation numbers throughout the paper. There are also many type errors throughout the text.

Another major flaw of the manuscript is that interpretations are not always fully supported by the data presented, and seem to be rather driven based on what the authors conventionally would expect based on common knowledge and previous research. For example, the interpretations for bands in the FTIR spectrum are not all convincing. In particular the peaks for treated NMPCB do not always match the lines or the bands that are indicated in the text.

Moreover, the poor quality of the figures does not help, or the images (in particular SEM) do not present evidence of what is interpreted. For example, Figure 4 c1 has a very light area that suggests charging on the surface, leading to very poor image quality. The samples should be properly coated to obtain good image quality.

Round 2

Reviewer 1 Report

The manuscript could be accepted for publication, the technical and scientific content has improved over the revision. However, editing and formatting still remain.

1. Proofreading by a native speaker or an editing service is needed as the current version of the manuscript has grammatical errors throughout the document.

2. 'apparent bulk density 103 of 0.55 g ml-1 and less than 5 % of impurity' -> a reference should be added, or state that this characterization is provided by the supplier.

3. 'particle size [...] was less than 200 μm' -> This statement does not have any scientific meaning as 'less then 100 μm' could mean anything between 0 and 100. Be more precise in this sentence.

4. Figure captions should indicate what the error bars mean, and number of samples should be mentioned.

5. Most of the references are incomplete, formatting is inconsistent, has many errors. The authors need to consult the journal guideline and must provide a uniform style for all references and the list should be error-free. Random superscipts are added after the names, names are incorrectly abbreviated, page ranges are missing, and in many cases the given pages after 'p.' are random numbers. This needs serious work.

Reviewer 3 Report

After the author's revision, I think this manuscript is basically ready for publication.

But the aesthetics of the fiugres in the manuscript still has great problems, especially in figures 2, 5, 6, 7, 8

Round 3

Reviewer 1 Report

Can be accepted.